# The Role of Lipids in CRAC Channel Function

**DOI:** 10.3390/biom12030352

**Published:** 2022-02-23

**Authors:** Lena Maltan, Ana-Marija Andova, Isabella Derler

**Affiliations:** Institute of Biophysics, JKU Life Science Center, Johannes Kepler University Linz, AT-4020 Linz, Austria; lena.maltan@jku.at (L.M.); amandova7@gmail.com (A.-M.A.)

**Keywords:** CRAC channel, STIM1, Orai1, protein-lipid interactions, modulatory proteins, ER-PM junctions, lipids

## Abstract

The composition and dynamics of the lipid membrane define the physical properties of the bilayer and consequently affect the function of the incorporated membrane transporters, which also applies for the prominent Ca^2+^ release-activated Ca^2+^ ion channel (CRAC). This channel is activated by receptor-induced Ca^2+^ store depletion of the endoplasmic reticulum (ER) and consists of two transmembrane proteins, STIM1 and Orai1. STIM1 is anchored in the ER membrane and senses changes in the ER luminal Ca^2+^ concentration. Orai1 is the Ca^2+^-selective, pore-forming CRAC channel component located in the plasma membrane (PM). Ca^2+^ store-depletion of the ER triggers activation of STIM1 proteins, which subsequently leads to a conformational change and oligomerization of STIM1 and its coupling to as well as activation of Orai1 channels at the ER-PM contact sites. Although STIM1 and Orai1 are sufficient for CRAC channel activation, their efficient activation and deactivation is fine-tuned by a variety of lipids and lipid- and/or ER-PM junction-dependent accessory proteins. The underlying mechanisms for lipid-mediated CRAC channel modulation as well as the still open questions, are presented in this review.

## 1. Introduction

Biomembranes play an important role in maintaining healthy cell function. They separate the interior from the exterior of the cell and form various compartments within the cytosol that perform a variety of cellular functions. Cell membranes are semipermeable, act as a diffusion barrier, and maintain a concentration gradient between different cell compartments. They are further involved in the regulated exchange of substances and control various cellular and intracellular processes like cell movement and subcellular signal transmission. The cell membrane is composed of two layers, namely the inner and outer leaflets which are made up of a variety of amphiphilic lipids. The hydrophobic tails of the lipids in both leaflets face each other while the hydrophilic headgroups meet the aqueous surroundings [1,2,3,4,5]. The lipid bilayer has an overall thickness of ~4 nm, is fluid, and contains proteins arranged in a mosaic pattern described by the fluid mosaic model [6,7,8].

Membrane lipids are categorized into glycerophospholipids, sphingolipids, and sterols [4,9]. The lipids of the first two groups are bilayer forming lipids. Although their overall structure, consisting of a hydrophilic headgroup linked to a hydrophobic tail, is similar, they may differ in fatty acids (chain length, presence of saturated state, and hydroxylation) and/or headgroup substituents, which leads to the existence of more than 1000 different lipid species [5,9]. Sterols are structurally different from the other two groups. They are composed of four aromatic rings, derived from sterane, and a short aliphatic chain, which makes them very rigid. It is noteworthy that sterols occur in mammals typically as cholesterol [4,10,11,12].

Given the diversity of lipids and the composition of lipid bilayer, it is hardly surprising that they influence not only the membrane properties, but also the structure and function of membrane proteins. Direct interactions between proteins and lipids or nonspecific changes in the physicochemical properties of the membrane, including curvature, fluidity, and thickness, can influence protein dynamics. In the case of lipid–protein interplay, a distinction can be made between annular and non-annular binding sites. Annular binding sites show direct interactions between lipids and membrane proteins, such as lipids located in the first transmembrane ring surrounding the protein. Non-annular lipids are located, for example, in crevices or at the interface of protein subunits [13,14,15,16]. Additionally, evidence is accumulating that membrane proteins are localized in micro- or nanodomains enriched in sphingolipids and cholesterol. These lipids facilitate the association of many signaling molecules, serve as a sorting platform for signal transduction proteins, and as a source of efficient signal transduction [17,18,19].

In regard to ion channels, it has been documented for decades that they can sense changes in the compositions of the lipid bilayer [20]. In particular, the so-called mechanosensitive channels are sensitive to changes in membrane pressure and the resulting change in membrane curvature and thickness. Examples of these types of ion channels include the bacterial large-conductance mechanosensitive channel (MscL) [21], piezo channels [22], and the potassium-selective TRAAK channel [23]. More recent reports concentrated on the role of signaling lipids, such as phosphatidylinositol-4,5-bisphosphate (PI(4,5)P2, short: PIP_2_) or diacylglycerol (DAG) in the direct modulation of ion channels. A prime example of ion channel gating by direct lipid binding is the inward rectifying K^+^ (Kir) channel. The direct binding of PIP_2_ to the transmembrane (TM) region 1 and 2 of all four subunits can trigger a conformational change and subsequently open the channel [24]. Moreover, some voltage-gated K^+^ (Kv) ion channels can function in a PIP_2_ dependent manner. Furthermore, a variety of ion channels (e.g., inwardly-rectifying K^+^ channels, voltage-gated K^+^ channels, Ca^2+^ sensitive K^+^ channels, voltage-gated Na^+^ channels, N-type voltage-gated Ca^2+^ channels and volume-regulated anion channels) are sensitive to changes in the cholesterol content of membranes [25,26,27,28]. Further examples of lipid-regulated ion channels include several members of the transient receptor potential (TRP) channels that can interact with different lipids at different channel positions (e.g., TRPV1/5/6 and TRPM2/8 interact with PIP_2_ [29,30,31,32,33,34] whereas TRPC3/6 are activated by DAG binding [35,36,37]). Lipids also affect channel activity by modulating their trafficking towards and insertion into the plasma membrane [38,39].

Pathological changes in lipid levels can either enhance or impair channel function, which may also play a crucial role in disease progression. In patients with hypercholesterolemia, the development of atherosclerosis or impaired vasodilatation has been associated with dysfunction or suppression of Kir or Kv channels, probably due to their modulation by cholesterol [25,26,40]. Altered availability of PIP_2_ or mutations that alter the function of PIP_2_-dependent channels have also been reported to adversely affect blood flow control and vascular function [41]. However, the effects of lipids on ion channels are highly variable and there are few examples of pathologies in which lipids and ion channels are co-regulated [25,26,40,41]. Hence, a detailed mechanistic understanding of the co-regulation of lipids and ion channels is required. This review focuses on the so-called Ca^2+^ release-activated Ca^2+^ (CRAC) ion channel, a prominent Ca^2+^ entry pathway into the cell. It consists of two components, one anchored in the endoplasmic reticulum (ER) membrane and the second in the plasma membrane (PM). Before describing these two key proteins of the CRAC channel in detail, the next section discusses the lipid composition and dynamics of the ER and the PM in the following.

## 2. Lipid Compositions and Dynamics in the Membranes of the ER and the Cell

Interestingly, the lipid composition of the ER membrane and the PM differ quite substantially. Even though the ER is the main place in the cell for lipid synthesis, it is mainly composed of phospholipids and a low amount of cholesterol. Its membrane is loosely packed, allowing transport of synthesized lipids and proteins. In contrast, in the PM, sphingolipids and cholesterol are abundant, which are responsible for its increased stiffness and mechanical resistance [9,42]. Moreover, the lipids in the two leaflets in the ER membrane of eukaryotic cells are symmetrically distributed as opposed to the PM, where almost all major lipids exhibit an asymmetric distribution [9,43,44]. While the outer leaflet is primarily composed of phosphatidylcholine (PC) and sphingomyelin (SM), the cytosolic leaflet contains particularly phosphatidylethanolamine (PE) and the charged lipids phosphatidylinositol (PI) and phosphatidylserine (PS). This asymmetry has major functional implications, e.g., on vesicle formation or the development of local transmembrane potentials. Furthermore, active transporters particularly maintain a higher content of anionic lipid species in the inner leaflet, which is of physiological importance. In addition, lipidomic studies revealed asymmetric distribution of acyl chain structure properties [43,45,46].

Cholesterol is an important hydrophobic lipid in the hydrophobic layer of the membrane [47], which controls lipid organization and conformational ordering of fatty acid chains [48]. Cholesterol has been proposed to be dynamically distributed in the PM, which likely depends on the composition, concentration of phospholipids, and forces in the lipid bilayer. However, in this respect more detailed studies are still required [49]. All PM phospholipids and cholesterol can be synthesized in the ER, which are then transported to the membrane, as reviewed by van Meer et al. [9].

In the PM, and especially at the contact sites between the ER and the PM, lipids are subject to dynamics, which are controlled by means of the so-called phosphoinositide cycle (Figure 1). Although phosphoinositides (PIs) represent only a small fraction of the total lipids in the PM, they are essential players in cellular processes such as actin dynamics, transmembrane protein regulation, and signal transduction [50,51,52,53,54,55,56,57]. PIs belong to the group of glycerophospholipids that contain inositol and carry different numbers of phosphate groups at their headgroups [58,59]. Specific kinases and phosphatases can phosphorylate the inositol ring at positions 3, 4, and/or 5, generating different subspecies that are physiologically relevant [58,59,60,61,62].

In the PM, PI species are phosphorylated by phosphatidylinositol-4-kinase (PI4K) to generate phosphatidylinositol-4-phosphate (PI4P). Consequently, PI(4,5)P2 or short PIP_2_ is produced by phosphorylation of phosphatidylinositol-4-phosphate-5-kinase (PIP5K) [60,61,63,64]. It is worth noting that specific counterpart phosphatases, as for instance the inositol polyphosphate 5-phosphatases (IPP-5Ptases) or the PIP_2_-4 phosphatases (PIP_2_-4Ptases), dephosphorylate PIP_2_ to PI4P and PI5P, respectively [65]. PIP_2_ can be further phosphorylated to PIP_3_ or reconverted by corresponding other kinases or phosphatases, respectively. Within the PI cycle, PIP_2_ is then hydrolyzed by phospholipase C (PLC) to diacylglycerol (DAG) and inositol-1,4,5-trisphosphate (IP_3_) (Figure 1). Following this, DAG activates protein kinase C (PKC), while IP_3_ binds its corresponding receptor in the ER membrane and activates Ca^2+^ release (Figure 1). When the ER membrane and the PM are in close contact, DAG is transported via non-vesicular transport from the PM to the ER membrane. Subsequently, DAG is sequentially converted by the diacylglycerol kinase (DGK) to phosphatic acid (PA), by CDP-diacylglycerol synthase (CDS) to CDP-DG and by the PI synthase (PIS) to PI. Finally, PI is transferred from the ER membrane to the PM via non-vesicular phosphatidylinositol transfer protein (PITP), which closes the circle and makes further rounds in the PI cycle possible (Figure 1). Furthermore, PA can be converted in the PM from DAG, which is then transported by PITPs to the ER membrane, where it is converted back to PI and transported back to the PM [60,61,63,64]. In this way, the spatiotemporal modulation of certain phosphoinositide lipid species at the ER-PM junction is critical for the control of Ca^2+^ signaling [13], as specifically outlined in detail for the prominent Ca^2+^ release-activated Ca^2+^ (CRAC) ion channel in the following sections.

## 3. CRAC Channels

As mentioned earlier, the CRAC ion channel consists of Orai, a hexameric plasma membrane protein that forms the pore unit [66,67], and the stromal interaction molecule (STIM), which protrudes from the ER membrane. While the Orai family consists of Orai1, Orai2, and Orai3, the STIM proteins include two members STIM1 and STIM2. Orai isoforms can form heteromeric channels and both STIM1 and STIM2 can interact with Orai1. Therefore, physiological CRAC channels are highly diverse [68,69,70,71,72]. In this review we focus on STIM1 and Orai1, as lipid-mediated regulation has so far been reported exclusively for those two proteins. However, given the structural homology of Orai and STIM isoforms, it cannot be precluded that they interact with lipids.

Upon store-depletion STIM1 can interact cytosolically with Orai1 after its activation [73,74] (Figure 2). Their interplay is therefore only given when the two membranes are in close proximity with each other.

STIM1 exists in the form of a dimer in the quiescent state and senses the luminal Ca^2+^ concentration in the ER via its N-terminal EF-hand [75,76,77]. This Ca^2+^ sensing region is connected via a TM domain with the C-terminus, which is involved in the control of the inactive state in the resting state and in the oligomerization of STIM1 and coupling to Orai1 in the activated state. The C-terminus exhibits a folded configuration in the resting STIM1 state in which CC2 and CC3 resemble the capital letter “R” and form an antiparallel V-shape in the dimeric state [78]. In this quiescent state, this V-shape is upside down and faces the ER membrane. A drop in the Ca^2+^ concentration leads to loss of the bound Ca^2+^ and subsequently to a huge conformational change of STIM1 [79,80,81,82]. This change starts with a slight unfolding of the ER luminal EF-hand and the sterile alpha motif (SAM) domain [76,77]. Following this, the opening signal is transduced via the TM domain to the three coiled-coil domains (CC1-3). In the activated state, CC1 is first zipped together, causing the CC2/CC3-V-shape to flip upward facing now the plasma membrane [83,84]. Here, the STIM1 apex (aa 390–410) together with a region N-terminal to the apex (aa 370–380; [85]) is now able to interact with Orai1, with the Orai1 C-terminus acting as the main binding site (Figure 2).

Several studies identified different fragments (OASF (aa 233–474) [86], SOAR (aa 344–442) [87], CAD (aa 342–448) [88] or Ccb9 (aa 339–444) [89]) of the CC domains which are sufficient to activate Orai1. The NMR structure of STIM1 and Orai1 C-terminal fragments, specifically the STIM1-Orai1 association pocket (SOAP), resolved the key sites required for the formation of the main STIM1/Orai1 binding site, which is in line with the findings of a variety of functional studies [90,91,92]. Recently, the STIM1 apex was also shown to be important in the intact communication with Orai1. In particular, position F394 in the STIM1 apex is a critical site of the protein [93], as mutations led to versatile defects ranging from the maintenance of the quiescent state (STIM1 F394A) to inability to oligomerize (STIM1 F394D/K) to impaired Orai1 binding capability (STIM1 F394H) [83]. Furthermore, cysteine crosslinking revealed that the STIM1 apex is essential for the interaction with the Orai1 loop2 [94]. Orai1 is composed of six subunits, each consisting of four transmembrane domains [95]. Two of the connecting loop regions extend into the extracellular matrix, while the third one as well as the N-/C-terminus protrude into the cytosol [66]. The Ca^2+^ selective pore of Orai1 is formed by the six TM1 domains which are surrounded by a second ring-like structure formed by TM2/TM3 and the outmost ring formed by TM4. Both TM1 and TM4 have longer overhanging helical segments extending into the cytosol, whereas for TM1 it is the extended transmembrane Orai1 N-terminal (ETON) region (20 Å) [96] and TM4 is connected by the hinge region to the helical C-terminus [97]. The high Ca^2+^ selectivity is mainly determined by the properties of the pore as well as the interaction with STIM1. The pore consists of a Ca^2+^-accumulating region with negatively charged amino acids (Orai1 D110, D112, and D114) that attracts Ca^2+^ ions at the extracellular side [98]. This is followed by Orai1 E106, the selectivity filter and narrowest part of the pore [95], a hydrophobic section (Orai1 L95, F99, and V102) [99], and finally the ETON regions with positively charged amino acids (Orai1 K83, K85, and R91) that repel ions in the resting state of Orai1 [100]. Mutations in all of these regions have severe effects on the normal function of Orai1 (e.g., Orai1 E106Q blocks Ca^2+^ permeation [101], Orai1 V102A impairs selectivity that can be rescued by STIM1 binding [96,102], and Orai1 R91W is one of the best-known loss-of-function (LoF) mutations leading to severe combined immune deficiency [66,103]).

However, it is not only the pore unit that is critical for channel activation; rather, a very large number of positions in all TM domains of the channel complex must be in an opening-permissive state to ensure an ideal working process of the CRAC channel complex. Indeed, a single point mutation at more than a dozen different positions in the four TM domains can result in either gain- or loss-of-function (GoF or LoF), depending on the properties of the incorporated amino acid [104,105,106,107,108,109,110,111]. Because an LoF mutation always acted dominant over a GoF mutation in most of the Orai1 mutants tested, it was demonstrated that Orai1 pore opening requires a global conformational change of the entire channel complex [108,112,113,114] in line with recent structural resolutions of an Orai1 mutant in the open state [95]. 

STIM1 and Orai1 are sufficient to constitute the CRAC channel [115]. However, due to the location of STIM1 and Orai1 in the ER membrane and the PM, respectively, and their assembly at ER-PM junctions upon store-operated activation, it is not surprising that their function is modulated by lipids. Furthermore, there is increasing evidence that cholesterol-rich regions define the association and function of the STIM1/Orai1 complex [116,117]. On the one hand, the function of both STIM1 and Orai1 is modulated by direct interaction with lipids, including particularly cholesterol and the phospholipid PIP_2_ [118,119,120,121,122,123]. On the other hand, there is plethora of evidence that the STIM1/Orai1 interplay is indirectly controlled via the spatially separated phospholipid bilayers, the ER and the PM, along with a variety of lipid- or ER-PM junction-dependent accessory proteins [124]. These forms of direct and indirect lipid-mediated modulation of CRAC channel function will be reviewed in the following.

## 4. Lipids Directly Regulated CRAC Channel Components

The major lipids which regulate STIM1/Orai1 function via direct interaction are PIP_2_ and cholesterol [25,125,126,127]. Furthermore, there is evidence for sphingomyelin- [128] and S-acylation- [123] mediated modulation of CRAC channels (Figure 2). Phospholipids were hypothesized to contribute to LoF of certain STIM1 mutants [83].

### 4.1. Phosphoinositides

Shortly after the discovery of STIM proteins, PIP_2_ was reported to be critical for the co-regulation of STIM1 and Orai1. In particular, reducing PIP_2_ levels using an inositol 5-phosphatase decreased store-operated Ca^2+^ entry. Both the STIM1 C-terminus and Orai1

N-terminus were shown to be sensitive to alterations in PIP_2_ levels [129,130] (Figure 2, Table 1).

At the very end of the C-terminus of STIM proteins is a lysine-rich region that functions as a PIP_2_ binding site in both STIM1 and STIM2 [88,179,180,181,182] (Figure 2). Mutation of these positively charged residues or deletion of the polybasic region impairs the stable association of STIM proteins at the PM after store-depletion. The stable coupling of STIM to the PM is promoted by the presence of PIP_2_ and also PIP_3_ in the PM, which are mainly enriched in cholesterol-rich regions [117,130,141,179,183,184]. Strikingly, the association of STIM1 with PIP_2_ necessitates tetramerization of its lysine-rich region, whereas efficient binding of STIM2 to PIP_2_ depends on dimerization of this polybasic domain. This leads to an enhanced affinity for PIP_2_ and a reduced activation threshold of STIM2 [185]. After deletion of this polybasic motif, ER depletion-induced translocation of STIM1 to the PM is impaired, even though STIM1 retains its ability to form oligomers [179]. In fact, although the lysine-rich region is dispensable when both STIM1 and Orai1 are overexpressed, it enhances the efficacy of the STIM1/Orai1 activation process under physiological conditions. STIM1 first assembles with PM-localized PIP_2_ and PIP_3_ pools in cholesterol-rich regions before interacting directly with Orai1 [186] (Figure 2). This suggests that cholesterol indirectly drives the accurate STIM1-mediated targeting of SOCE components in lipid rafts by retaining the necessary phosphoinositide pools.

The N-terminus of Orai1 contains a polybasic region that was reported to be sensitive to PIP_2_ levels in the PM and essential for directing Orai1 to membrane areas with distinct PIP_2_ content [129]. However, the dependence of Orai1 on STIM1 makes it currently challenging to characterize PIP_2_ regulation of Orai1 channels alone. One potential way to determine whether Orai1 is directly regulated by PIP_2_ is to investigate the effect of PIP_2_ depletion on one of the various, currently known, constitutively active Orai1 mutants, which were reported for instance in Tiffner et al. [108] and Yeung et al. [104].

Interestingly, PI4P, the precursor of PIP_2_, was also reported to modulate CRAC channel function [130,131,132] (Figure 2). The phosphatidylinositol 3- and phosphatidyl-inositol 4-kinases inhibitor, Y294002, resulted in inhibition of store-operated Ca^2+^ entry in a concentration-dependent manner in human platelets [130,131]. The PI4K inhibitor, wortmannin, also led to inhibition of store-operated currents, even though STIM1 puncta formation was preserved [130,132]. Furthermore, knock-down of the PI4K reduced endogenous SOCE [130]. To further investigate the role of PI4K, the newly synthesized and more specific PI4P inhibitors might be highly valuable [133].

Overall, PIP_2_ and PI4P represent modulatory factors of the CRAC channel machinery (Figure 2). While it is clear that STIM1 interacts directly with PIP_2_, the effects of PIP_2_ alone on Orai1 are less explored. Further studies are still required to understand the role of PI4P levels in the CRAC channel choreography.

### 4.2. Cholesterol

Cholesterol was shown to modulate the function of both STIM1 and Orai1, likely via direct interaction [120,121,122] (Figure 2, Table 1).

Very early studies showed that the removal of cholesterol from membranes reduced SOCE and decreased the ability of a constitutively active STIM1 mutant (D76A) to form punctae [135]. However, there are conflicting studies on the effect of depletion of cholesterol by methyl-β-cyclodextrin (MßCD) on CRAC influx. SOCE in cells with endogenously expressed STIM1 and Orai1 was reduced by the application of MßCD [134]. In another study, it was demonstrated that cholesterol depletion mediated by cyclodextrin led to Orai1 internalization and altered Orai1 diffusion, which was restored by Cav-1 overexpression. While cholesterol depletion by MßCD reduced STIM1-mediated Orai1 currents, co-expression of Cav-1 maintained or even enhanced current levels observed in the absence of MßCD application as well as Cav-1 overexpression [122]. In contrast, Gwozdz et al. [119] demonstrated that overexpression of STIM1 and Orai1 abolished the inhibitory effect of chemical cholesterol depletion by MßCD on SOCE.

We found that depletion of cholesterol via cholesterol oxidase or filipin enhanced SOCE in both human embryonic kidney 293 (HEK 293) cells and rat basophilic leukemia 2H3 (RBL) cells [120]. Similarly, Pacheco et al. [121] demonstrated that cholesterol depletion by MßCD enhanced Orai1 currents only after STIM1- or SOAR-Orai1 coupling [121]. Interestingly, MßCD treatment leads to enhanced STIM1-Orai1 coupling [121], whereas the application of cholesterol oxidase left STIM1-Orai1 coupling unaffected [120]. The observed differences are likely due to the gentler manipulation of cholesterol levels by cholesterol oxidase or filipin compared with MßCD, while leaving membrane integrity intact. The differences in membrane composition might also be explained by the distinct effects on STIM1/Orai1 coupling upon the different ways of chemical cholesterol depletion. Furthermore, MßCD treatment was shown to decrease membrane localization of STIM1 and disrupts its interaction with Orai1 [129,134]. Alternatively, distinct effects could occur upon overexpression toward endogenous proteins [119].

Observed effects due to chemical cholesterol depletion underlie the direct interplay of cholesterol with STIM1 as well as Orai1 [120,121].

Indeed, we showed direct binding of TopFluor-cholesterol to Orai1 [120] (Figure 2). Searching for a potential cholesterol binding site, we discovered the so-called cholesterol recognition amino acid consensus motif (-L/V-(X)(1-5)-Y-(X)(1-5)-R/K-; X represents from 1 to 5 any amino acids before the next conserved residue) [11,187,188,189] in the ETON region (aa 73–90), which is integral to channel gating by STIM1 [120]. Indeed, mutation of critical sites in this cholesterol binding motif (L74I, Y80S) led to reduced interaction of cholesterol, with both full-length Orai1 mutants as well as an Orai1 N-terminal peptide [120] mutated accordingly. In line with the enhanced currents after chemical cholesterol depletion, Orai1 mutants deficient in cholesterol binding also showed an increase in SOCE without affecting the interaction with STIM1. This indicates that STIM1-Orai1 interaction at the C-terminus can still occur, with cholesterol only interfering with STIM1-dependent channel gating through Orai1 N-terminal interactions. However, it is still a matter of debate whether this cytosolic cholesterol binding motif directly interacts with cholesterol or allosterically affects cholesterol binding to Orai1. Based on the Orai channel structures, the proposed N-terminal cholesterol binding site contains the critical positions L74 and Y80. However, they point in opposite directions, which is unfavorable for cholesterol binding. For direct Orai1-N-terminus-cholesterol interaction to take place, the ETON domain would need to be in equilibrium between the α-helix (as in the crystal structure) and random coil structures, as possibly occurring in the isolated Orai1 N-terminal peptide in solution [120]. In this case, cholesterol would most likely favor a random coil structure and make Orai1 unavailable for STIM1 interaction. However, it seems unlikely that a stable α-helix would “unwind” under physiological conditions. Therefore, a more energetically favorable possibility would be a conformational change between an α-helix and the rarer 3_10_-helix structure (with three residues per turn as opposed to the 3.6 residues per turn in a canonical α-helix) [190]. This allows L74 and Y80 to align in the same direction and enable cholesterol binding. Alternatively, it could be hypothesized that the direct cholesterol binding site is located at a distinct site within the Orai1 channel complex which interplays allosterically with the N-terminus, but this requires further study.

Cholesterol associated effects were also linked to a cholesterol binding motif with the consensus sequence L/V-X(1–5)-Y-X(1–5)-R/K in the STIM1 C-terminus [121]. Indeed, a mutation within this binding motif (I364A), located in SOAR, resulted in comparable enhancements of Orai1 currents to chemical cholesterol depletion in both full-length STIM1 and a C-terminal STIM1 fragment. Via MD simulations, it was confirmed that cholesterol affects the coupling of SOAR to the membrane, with I364 being the major interface. A combination of Orai1 and STIM1 mutants, both deficient in cholesterol binding, did not further enhance currents compared with ones obtained by mutating either STIM1 or Orai1. This indicates that removal of the cholesterol-binding (CB) domain of either Orai1 or STIM1 is sufficient to enhance calcium influx and Orai1 currents, mimicking the effect of cholesterol removal from the PM. Altogether, these results strongly suggest that the CB domain of both, Orai1 and STIM1 coordinates the same cholesterol-mediated mechanism [121].

Regulation of STIM1 or Orai1 by cholesterol may possibly have an underappreciated pathophysiological relevance. We suspected a connection between hypocholesterolemia and enhanced mast cell degranulation. Indeed, patients suffering from hypocholesterolemia tend to have an enhanced allergy response [191], in line with the findings that cholesterol depletion in RBL mast cells enhanced store-operated Ca^2+^ currents and degranulation [120]. However, further studies are still required in this regard. Moreover, the question arises whether other pathophysiological conditions are caused by the interaction of cholesterol and Orai1.

The relationship between cholesterol and Orai remains controversial in terms of its dynamics and physiological significance. Do ER-PM transitions form exclusively at cholesterol-rich regions, with cholesterol inhibition being the standard for “wild-type” SOCE reactions, or is cholesterol binding to Orai turned on and off? What conditions lead to the transport of cholesterol from the PM to its cytosolic binding site at Orai? Moreover, it is currently contentious whether STIM1 couples not only to Orai1 C-terminus, but eventually also to Orai1 loop2 or N-terminus [108,111,112,192,193,194,195]. Hence, how do cholesterol and STIM1 compete for the same region of Orai?

It could be hypothesized that when stores are depleted, STIM1 binds to the C-terminus of Orai1 and triggers a conformational rearrangement that removes cholesterol from the N-terminus and instead allows STIM1 to bind, causing the channel to open. This is an intriguing concept of cholesterol-mediated SOCE regulation (Figure 2). However, the physiological ramifications and the mechanism of when and why cholesterol binds to Orai1 remain to be elucidated. A detailed characterization of the Orai1 cholesterol binding pocket within the CRAC channel complex, even at the structural level, would be highly valuable for a more detailed understanding of the cholesterol-dependent molecular mechanisms.

### 4.3. Sphingomyelin

Sphingomyelin is very abundant in the PM of mammalian cells, accounting for about 20% of the total phospholipids in this bilayer, and is preferentially located in the outer leaflet [9]. The exchange of sphingomyelin with the inner leaflet is less likely due to its polar headgroup. Moreover, it is firmly anchored in the membrane by its hydrophobic part. Physiologically, sphingomyelin in the PM is essential for lateral homogeneity and acts as a source of lipid signaling [44,196,197].

A central position in the sphingomyelin metabolism is occupied by ceramide, which is produced by several enzymatic cycles in the ER. To form sphingomyelin, ceramide is transported to the Golgi. There, ceramides can be endowed with different headgroups resulting in the generation of distinct groups of complex sphingolipids, one of which is sphingomyelin. Sphingomyelin further acts as a substrate for ceramide phosphates and sphingosine, which are produced by specific enzymes. For instance, sphingomyelinase (SMase) D converts sphingomyelin into ceramide-1-phosphatase [196,198,199].

It was reported that sphingomyelin itself directly affects various ion channels [200,201,202]. Among these channels, the CRAC channel was also found to be a target of sphingomyelin (Figure 2, Table 1). SMase D treatment significantly reduced store-operated currents without affecting Ca^2+^ store-depletion. Although the application of SMase D can lead to the generation of high levels of ceramide phosphates, application of ceramide-1-phosphate did not reduce store-operated currents. These results indicate that CRAC channel modulation occurs directly via sphingomyelin and not via one of its metabolites [128] (Figure 2).

One plausible mechanism for the modulatory role of sphingomyelin on the CRAC channels is their direct interaction. However, further studies resolving their direct interaction are still required. Alternatively, SMase D treatment may change PM organization, thus altering CRAC channel activation [128,196].

### 4.4. S-Acylation

S-acylation, also called S-palmitoylation, is the posttranslational tethering of a medium length fatty acid, also called palmitic acid, to a cysteine residue. Importantly, unlike irreversible posttranslational lipid modifications such as prenylation and myristoylation, S-acylation of proteins is reversible. Thus, S-acylation allows dynamic and spatiotemporal control of protein activity and interaction [203,204,205]. For more than 50 ion channels, S-acylation was reported to control their gating and trafficking by enhancing the hydrophobicity of protein segments [205]. The S-palmitoylation reaction is mediated by protein acyl transferases (PATs) in the ER and Golgi containing zinc finger and DHHC domains [206]. The process is reversed in the PM by acyl protein thioesterases [207], among which 23 PATs and five thioesterase isoforms were so far identified in humans [208]. It is suggested that the hydrophobic properties of the attached acyl groups may affect the distribution of proteins in the membrane [209].

A recent report revealed the first indication that Orai1 activity is controlled by S-acylation. Orai1 was found to be S-acylated by ZDHHC20 (PAT20) at residue C143 (Figure 2, Table 1), which is critical in controlling its trafficking and function and ensures its localization in cholesterol-rich regions. Upon mutation of this cysteine, Orai1 was recruited into cholesterol-poor regions causing STIM1 mediated Orai1 currents to diminish and the signaling at the immunological synapse between T-cell and antigen presenting cell to occur less efficiently. Furthermore, downstream signaling events including long-lasting Ca^2+^ level enhancements, nuclear factor activated T-cell (NFAT) translocation and IL-2 secretion were abolished [123].

At this point, it is unclear whether S-acylation is required for or strengthens Orai1-cholesterol interactions. In this regard, the reduction in STIM1/Orai1 currents with impaired S-acylation are in line with the effects observed upon chemical cholesterol depletion by MßCD [119,134,135] but are at odds with store-operated current enhancements upon chemical cholesterol depletion by MßCD [120,121]. Therefore, it would be of interest to investigate whether Orai1 C143A is deficient in cholesterol binding. A potential cholesterol binding site was identified in the Orai1 N-terminus at positions located in the same plane such as C143. Thus, if Orai1 C143A is deficient in interaction with cholesterol, one could further test for a potential allosteric interplay of C143 with the N-terminal positions L74 and Y80 with respect to cholesterol binding.

### 4.5. Potential Role of Phospholipids in Controlling STIM1 Function

As part of our characterization of the role of a small alpha helical region within the CAD/SOAR apex of the resting STIM1, namely the α2 region, we identified distinct point mutations therein that can impact the STIM1/Orai1 activation cascade in various manners. STIM1 homomerization and consequent Orai1 activation was impaired upon single point mutation to hydrophilic, charged amino acids (STIM1 F394D, STIM1 F394K). Using MD simulations, we found that their loss of homomerization could be due to possible electrostatic interactions with lipid headgroups in the ER membrane and an altered formation of the CC1α1-SOAR/CAD interaction site. Consistent with these results, we demonstrated experimentally that the disruptive effects of F394D depend on the distance of the apex from the ER membrane (Figure 2, Table 1). Collectively, our findings provide first evidence that the CAD/SOAR apex is in the immediate vicinity of the ER membrane in the STIM1 quiescent state [83] which is in line with the recent finding by van Dorp et al. [84]. At this point further investigations are required to determine whether the close proximity of STIM1 apex to the ER membrane in the closed state is predominantly stabilized by intra- and intermolecular interactions within the STIM1 proteins or whether ER membrane lipid interactions also contribute. In the latter case, determining which lipids might be involved would be challenging.

## 5. Indirect Control of STIM1/Orai1 Machinery at ER-PM Membrane Contact Sites

In the following parts of the review, the focus is laid on the ER-PM contact sites, as this is the interface for the STIM1/Orai1 interplay.

### 5.1. ER-PM Constact Sites and Methods for Their Characterization

Membrane contact sites (MCSs) are microdomains where the membranes of two organelles get in close proximity without fusion. It is becoming increasingly evident that MCSs are critical to coordinate a variety of physiological events including organelle dynamics, lipid exchange, Ca^2+^ signaling and cell survival [210,211,212,213,214,215,216,217]. MCSs of the membrane of intracellular Ca^2+^ stores such as the sarcoplasmic reticulum or ER membrane and the PM control for instance the excitation-contraction coupling in muscle cells and store-operated Ca^2+^ entry in non-excitable cells, respectively [213].

Multiple technologies, including electron microscopy and fluorescence microscopy, have significantly advanced our understanding of MCS. In this regard, electron microscopy (EM) is a “gold standard method”, providing a static snapshot of MCS architecture at the nanoscale, albeit under non-native conditions. In contrast, cryo-electron tomography (cryo-ET), which immobilizes non-crystalline samples and allows 3D imaging at a high resolution of 4–10 nm, enables the resolution of MCS in a near-native state [217,218].

To visualize MCS structures in living cells, multispectral fluorescence microscopy uses genetically encoded fluorescent proteins attached to MCS-resident proteins. To overcome the diffraction limit of confocal microscopy at or near the PM [219,220], TIRFM is used to selectively illuminate fluorophores at surface regions with wave penetration of approximately 100 nm into the sample. Therefore, the dynamics and kinetics of proteins residing at the ER-PM contact sites, such as STIM1 puncta formation, can be easily monitored within 1 min after Ca^2+^ store-depletion [73,74,175]. An even better resolution can be achieved with super-resolution microscopy such as photoactivated localization microscopy (PALM) [221] and STochastic Optical Reconstruction Microscopy (STORM) [222] in conjunction with fluorophore-labeled MCS-resident proteins.

To identify unknown binding partners within MCSs, the bimolecular fluorescence complementation (BiFC) method can be used. This attempt is based on fluorescent proteins (FP) split into two non-fluorescent parts, each of which can be fused to two proteins residing in opposing membranes. The chromophore fluorescence is recovered only when the FP fragments are close to each other (10–30 nm) [223], thus demonstrating that the two proteins are in close proximity and might interact. As such, BiFC can serve as a great tool for characterizing native MCS distribution and dynamics under different pathophysiological conditions.

Furthermore, markers that allow remote modulation of intramembrane binding and MCS assembly were developed. These include chemically or light-inducible modules that can modulate the properties of MCS [218], among which we focus on those critical to modulate ER-PM junctions.

An initial tool to gain a better understanding of STIM1-Orai1 communication at ER-PM junctions is the MAPPER (membrane-attached peripheral ER) [140,224]. This marker consists of a signal peptide of STIM1 located in the ER lumen followed by the TM domain of STIM1. Subsequently, an FKBP12-rapamycin binding (FRB) domain followed by the polybasic motif of the small G protein Rit, which binds PIP_2_ and PIP_3_ in the PM [182], were added. Several flexible and helical linkers were inserted in the cytosolic portion of MAPPER to guarantee that its expression did not change the gap distance at the ER-PM junctions. MAPPER localized in puncta at the ER-PM contact sites at analogue positions like activated STIM1. A light-sensitive variant of this is the LiMETER (light-inducible membrane tethered peripheral ER tool), based on a STIM1-Rit chimera [175]. Compared to the MAPPER, LiMETER contains the photosensitive LOV2 domain (aa 404–546) instead of the FRB domain. In the inactive state, LiMETER is tightly packed and hides the polybasic region. Upon activation by blue light, this is released, leads to clustering, and interacts with the PM. Remarkably, this process is reversible and therefore could be used to detect protein interactions, as performed with STIM1 and the accessory protein STIMATE [175] (see Section 5.4.5).

Other markers of ER-PM junctions are OptoPB and OptoPBer [225]. Like LiMETER, both contain the LOV2 domain and polybasic regions. The latter could be used for interaction studies with specific lipids, such as Rit1 and PIP_2_ [182]. While OptoPB is a cytosolic protein, OptoPBer is anchored in the ER membrane such as LiMETER. Thus, OptoPBer cannot only be used to study interactions of polybasic domains and lipids, but additionally serves as a spacer between the ER and PM [225]. Upon activation with blue light, OptoPBer forms puncta and reversibly reverts to a uniform distribution upon inactivation in the dark. These tools helped to provide novel insights into the CRAC channel machinery [175,218]. For example, to investigate key interaction residues of the polybasic region of STIM1 with the PM, He et al. [225] initially exchanged lysines with alanines within OptoPBer and utilized positive hits on full-length STIM1. They found that the STIM1 double mutant K684A K685A abrogates punctae formation [225] and may also be insufficient to activate Orai1. Furthermore, variation in the linker length between TM and LOV2 domain revealed that the working distance of the CRAC channel is between ~15–30 nm [225].

Taken together, these methods provide the opportunity to obtain a comprehensive picture of the interaction and distance between organelles and the cell membrane. In addition, the optogenetic tools allow precise temporal and spatial control over the formation of contacts between the membrane and organelles. This could also be used to reversibly switch the binding of peptides or proteins to the PM between near and far PM, which is a promising way to study protein–protein interactions. In this context, it should be noted that the genetically encoded photosensitive proteins are still in the development stage and many efforts are still being made to improve them. Their range of applications can be extended immensely by varying, for example, the frequency of the light pulse, the length of the linker between the fusion proteins or the expression ratio of the cellular components under investigation, and will thus be increasingly used as the methodology improves.

### 5.2. Critical Factors Determining STIM1/Orai1 Co-Regulation at the ER-PM Contact Sites

The ER-PM junctions are currently the best understood cellular MCSs. Typically, the ER-PM junctions make up only 0.8–2.5% percent of the cell surface. They are separated by a gap of 10–25 nm and have sizes of 100–200 nm [74,145,226,227]. ER-PM junctions are highly dynamic contact sites which depend on a variety of factors, including (i) spatial and temporal fluctuations of the lipid compositions regulated by the PI cycle as described in Section 2 [13], (ii) cytosolic Ca^2+^ concentration [74,213], as well as (iii) a set of proteins located at these contact sites forming interactions with proteins or lipids [13,118].

ER-PM contact sites are particularly specialized sites for Ca^2+^ signaling. This is not surprising since the ER functions as the largest Ca^2+^ store in the cell. Thus, Ca^2+^ plays an essential role in the formation of MCSs of various organelle membranes with the ER membrane [213]. With regard to ER-PM junctions, it was reported that the luminal Ca^2+^ levels affect their development and stability. While a certain number of ER-PM contact sites are already present in the resting state of the cell, Ca^2+^ store-depletion triggers the formation of new contacts [74,140]. These events are often driven by the signaling steps of the elementary unit of CRAC channels formed by STIM1 and Orai1, which accumulates in ER-PM contact sites upon store-depletion. The expression of STIM1 alone can already influence the formation and size of ER-PM junctions. Indeed, the overexpression of STIM1 greatly increased the dimensions of the ER-PM contact sites, whereas the role of endogenous STIM1 is still unclear [226,227,228]. In particular, store-depletion triggers STIM1 puncta formation at the ER-PM contact sites by binding to PIP_2_ in the PM. As a result, STIM1 couples to Orai1 through diffusional trapping. The interaction between STIM1 and Orai1 is the basic requirement for CRAC channel activation [73,115,229], but the local organization and dynamics of the membrane architecture at the ER-PM junction influence the efficacy of their interaction and gating [141,170]. The latter involves several accessory proteins that control lipid metabolism at and the formation of ER-PM contacts, some of which are also Ca^2+^ sensitive. They can be grouped into proteins either spanning from the ER to the PM or anchored in/at one of the two membranes [13], as described in the following.

### 5.3. ER-PM Spanning Proteins Involved in the Modulation of STIM1/Orai1 Function

Proteins which are located in the ER and can form direct or indirect contacts with the PM comprise the Extended-synaptotagmins (E-Syts) with E-Syt 1-3 [136], VAMP associated proteins (VAP) [230], GRAMD2A [159], and Anoctamine 8 (ANO8) [160] (Figure 3, Table 1).

#### 5.3.1. Extended Synaptotagmins

E-Syts (E-Syt1/2/3), which are the mammalian homologues of tricarbins (TcB1/2/2) in yeast [136,140], are integrated into the ER membrane and contain a cytosolic synaptotagmin-like mitochondrial lipid-binding protein (SMP) domain. Dimerization of the SMP domain allows E-Syts to form homo- and heteromers. Furthermore, the SMP regions enable non-vesicular lipid exchange, in particular of DAG from the PM to the ER (Figure 3). Downstream of the SMP domain, E-Syts contain multiple PIP_2_- (Figure 3) and Ca^2+^-binding C2 domains. [136,137,138]. E-Syt1 necessitates an enhancement in intracellular Ca^2+^ levels to interact with the PM and allow lipid transport. E-Syt2 and E-Syt3 interact constitutively with the PM independent of Ca^2+^ [136,139]. ER-located E-Syts establish the ER-PM contact sites by interacting with PIP_2_ in the PM [140,141,142].

Notably, the deletion of all three E-Syt proteins does not affect STIM1/Orai1 activation [136] and that E-Syt triple knockout mice have no apparent phenotype [231]. This suggests that the functionality of ER-PM junctions can be sustained by compensatory machineries, which likely include several other proteins involved in the ER-PM junction development, which likely include some of those described in the following subsections.

Nevertheless, E-Syts were found to control STIM1 function in a cell-type specific manner. In HeLa cells, they were shown not to be important for store-operated Ca^2+^ entry. Downregulation of E-Syts reduced the number of ER-PM junctions, but did not reduce SOCE. This suggested that STIM1-Orai1 contacts are formed at distinct sites than those of E-Syt proteins [136]. Later, E-Syt1 was reported to be activated in HEK293 cells after store-operated Ca^2+^ entry. However, E-Syt1 did not contribute to the formation of ER-PM junctions supporting the interaction of STIM1 and Orai1, but rather rearranged adjacent MCSs which facilitated the repletion of ER stores with Ca^2+^, thus supporting the deactivation of STIM1 [142]. In T-cells, E-Syt1 and E-Syt2S, a short isoform of E-Syt2, contribute to CRAC channel activation as their knockdown decreased STIM1 clustering, SOCE as well as cytokine release. The reason for the difference in HeLa and T-cells is that E-Syt2S occurs predominantly in T-cells and recruits STIM1 to ER-PM contact sites via direct interactions. In fact, while deletion of the STIM1 PIP_2_ binding site, specifically the lysine-rich region, at the very end of the STIM1 C-terminus (STIM1 ΔK) was unable to translocate into puncta in the absence of Orai1, only co-expression with E-Syt2S triggered STIM1 ΔK oligomerization. The long isoform of the E-Syt2 protein, E-Syt2L, is assumed to act inhibitory on STIM1 recruitment to the PM [143] (Table 1).

#### 5.3.2. VAP Proteins

VAPs (VAP-A, VAP-B), which are homologues to yeast Scs2p/22p, do not make direct contacts with opposing membranes but form multi-subunit bridges. They consist of a C-terminal TM domain located in the ER, which is connected to a linker of 26–27 nm length followed by a segment which contains the binding site for proteins possessing the FFAT (“two phenylalanine in an acidic tract”) [147,148,149]. VAPs interact with various lipid transport proteins, which contain an FFAT binding site (Figure 3) to form contacts with the PM.

One such protein linked to lipid transfer at ER-PM contact sites and essential in the STIM1/Orai1 machinery is the Nir2 protein, also known as PYK2 N-terminal domain-interacting receptor (Nir) 2 because of its binding to PYK2-binding proteins. It is a member of the PI transfer protein (PITP) family and dynamically recruited to ER-PM junctions on the one hand by the interaction with the ER-localized VAP-A and VAP-B proteins [230] and on the other hand by binding to DAG and/or PA in the PM [140,232,233]. It is responsible for the exchange of PI and PA between the two membranes of the ER-PM junctions [118,232,234] (Figure 3, Table 1). Moreover, it regulates PIP_2_-Ca^2+^ signaling and maintains PIP_2_ levels during PLC activity. Indeed, downregulation of Nir2 diminished PIP_2_ replenishment [118,140,224,232,233,234]. In addition to VAP proteins, the recruitment and function of Nir2 is supported by E-Syt1 as it was shown by the ER-PM marker MAPPER. This suggests a functional interplay between E-Syt1 and Nir2 [140]. Nir2 stabilization is further facilitated by their interaction with F-actin at the ER–PM contact sites [145].

Ca^2+^ store-depletion triggers E-Syt1 translocation to the ER-PM contact sites to trigger the formation of new ones and reduce the distance between the ER and the PM. This facilitates the recruitment of Nir2 to the PM and the enhanced transfer of phosphatidylinositol (PI) to the PM which is converted to PIP_2_ to support Ca^2+^ signaling initiated via receptor stimulation [146,232,235] (Figure 3). In this manner, Ca^2+^-dependent accumulation of E-Syt1 may help maintain PIP_2_ levels in local areas while DAG is shuttled to the ER to impede its accumulation in the PM (Figure 3) [140,144]. This likely supports the observed colocalization of both E-Syt1 and Nir2 with STIM1 within the ER-PM junctions to prime the cell for additional stimulation [176,236].

Furthermore, among oxysterol-binding proteins (ORP), ORP3 was reported to be critical in STIM1-Orai1 signaling. ORPs are a family of lipid transfer proteins containing twelve human members. In particular, ORP3, ORP5, ORP6, and ORP8 perform lipid exchange functions at ER-PM contact sites [152]. ORP3 and ORP6 bind to VAP proteins via their FFAT binding site [148,150] (Figure 3, Table 1). Together with ORP5 and ORP8, which are embedded in the ER membrane via their C-terminus and bound to PM phosphoinositides via their PH domain [148,150,153] (Figure 3), they allow non-vesicular lipid transfer between MCSs. Specifically, phosphatidylserine (PS) and sterols in the ER are exchanged by PI4P in the PM [237,238,239,240,241,242,243]. It is worth noting that ORP5 and ORP8 can bind either PS or PI4P, but not both simultaneously. PI4P is supplied by ORP proteins to Sac1 phosphatase in the ER (Figure 3). The latter, together with the PI4KA enzyme in the PM [133,244,245], are responsible for maintaining the PI4P gradients between the ER and PM. In this way, ORP proteins synergistically trigger the activation of PIP5K and consequently synthesis of PIP_2_ [237,246]. However, the question remains as to how the cell makes it possible that PI4P can be used for both, the production of PIP_2_ in the PM and the exchange of PI4P/PS. One possibility is the functional segregation of two pools of lipids [133,245,247] (Figure 3). In addition, it should be pointed out that PI4P metabolism has a great influence on PS content, so PS abundance could have a major impact on PI4P content in the PM [248,249].

ORP3 was reported to modulate STIM1/Orai1 function. It requires PKC activity and depends on PI4P and PIP_2_ levels to attach to the PM. Moreover, ORP3 translocation to the PM is triggered by store-operated Ca^2+^ influx. After activation, ORP3 mainly extracts PI4P from the PM. Complete activation of ORP3 led to a decrease in the level of PI4P in the PM and hampered Ca^2+^ influx via the store-operated Ca^2+^ entry pathway. The correct localization of ORP3 is determined by its C-terminus [150,151] (Table 1).

Since cholesterol [119,120,121,122] and PI4P [130,131,132] seem to play a role in STIM1/Orai1 signaling, continuing studies are required to determine whether the other ORP proteins are involved in the modulation of the CRAC channel function as well. In addition, Ca^2+^ entry into activated T-cells was reported to alter PS levels in the vicinity of T-cell receptors, possibly triggering their conformational changes [250]. Whether Ca^2+^ entry is also related to ORP-mediated alterations in the PS content in the vicinity of the STIM1-Orai1 complex and whether this has implications for the structure–function relationship of CRAC channels remains to be elucidated.

#### 5.3.3. GRAMD2a

Of the human GRAM proteins, GRAMD1a and GRAMD2a form ER-PM tethers, in analogy to their yeast orthologs Ltc/Lam proteins [157,158,251]. They are composed of a C-terminal TM domain anchored within the ER membrane. Downstream of the TM domain, they contain one or two START-like VASt domains that bind sterols and facilitate their transport, a PH-like GRAM domain that binds PIP_2_, and an unstructured N-terminal region [155,156,157,158] (Figure 3). They were reported to play a role in the sterol transport from the PM to the ER [157,158,252].

Among the different GRAMD proteins, GRAMD1a and GRAMD2a were recently reported to co-localize with STIM1 and E-Syt at the ER-PM contact site. Notably, PIP_2_ is required for this assembly. GRAMD2a influenced STIM1 puncta formation because knockdown of GRAMD2a reduced the area, but not the number, of STIM1 clusters activated upon store-depletion. Moreover, silencing GRAMD2a significantly altered E-Syt1 localization, whereas store-dependent Ca^2+^ entry was unaffected [159]. Hence, GRAM domain proteins potentially support STIM1 PM translocation, but they are not required for SOCE, and their functional role still needs to be clarified in more detail. For instance, typical CRAC channel hallmarks such as fast or slow Ca^2+^ dependent inactivation in the absence compared to the presence of STIM1 could be investigated to further characterize the role of GRAMD proteins in CRAC channel signaling. Nevertheless, GRAMD2a proteins definitely serve as alterative markers for ER-PM junctions (Table 1).

#### 5.3.4. ANO8

Proteins of the anoctamin family were recently suggested as the mammalian homologue of yeast Ist2 based on sequence analysis. However, the role in the function of the STIM1-Orai1 interplay and Ca^2+^ signaling events remained unknown [253,254]. The anoctamin family consists of ten proteins about whose properties and cellular functions very little is known, except for ANO1 [255,256,257], ANO2 [258], and ANO6 [259,260,261]. ANO1 and ANO2 are Ca^2+^-activated Cl^−^ channels [262], while other anoctamins were also reported to have some scramblase activity. ANO6 is a lipid scramblase that triggers the transport of PS and is supposed to function as Cl^−^ channel independent of Ca^2+^ [253,259]. Recently, ANO8 was reported to control the CRAC channel machinery. Akin to other ANO proteins and its yeast homologue Ist2, ANO8 was predicted to possess ten TM domains located in the ER membrane and to have a long cytosolic C-terminus containing a series of basic residues that likely form the putative PIP_2_ binding motifs (Figure 3).

Recently, the anoctamin ANO8 was reported to function as an essential ER-PM tether at PIP_2_-rich regions that controls the fundamental features of Ca^2+^ signaling. ANO8 regulates store-operated STIM1-STIM1 and STIM1-Orai1 coupling, STIM1 clustering, store-dependent Ca^2+^ influx, and Orai1 activation and subsequent inactivation. Elimination of the C-terminal PIP_2_ binding site in ANO8 abolished its promoting effects on the STIM1/Orai1 activation machinery. Moreover, ANO8 promoted Orai1 channel inactivation by substantially augmenting SERCA2-mediated Ca^2+^ entry into the ER, even at low cytosolic Ca^2+^ levels of 0.2 nM. One explanation for the observed effects is that ANO8 leads to an assembly of the core CRAC channel components and modulatory proteins (Orai1, PMCA, STIM1, IP_3_R, SERCA2) at PIP_2_-rich regions in the ER-PM junctions. Thereby, they are able to establish efficient Ca^2+^ signaling that governs all variations of receptor-evoked Ca^2+^ oscillations and the duration of Orai1 activity [160]. It is currently unclear whether ANO8 is more likely to strengthen ER-PM junctions, either directly through interactions with other ER-PM spanning proteins or indirectly. Alternatively, it is possible that ANO8 binds directly to one of the key players in the CRAC channel. However, further studies in this regard are still required (Table 1).

### 5.4. ER- or PM-Tethered Proteins Located in the ER-PM Junctions and Modulating the STIM1/Orai1 Interplay

In addition to the proteins that establish direct contact between the two membranes at the ER-PM junctions, STIM1/Orai1 activation at these membrane contact sites is controlled by other proteins located either in the ER membrane or in the PM. Some of them also regulate lipid dynamics in the membrane. Among the latter, we focus in particular on those proteins essential in the modulation of the STIM1/Orai1 complex and located at ER-PM contact sites in the following subsections. These include septins, RASSF4, STIMATE, SARAF, and Caveolin-1 (Cav-1) (Figure 4 and Figure 5, Table 1).

#### 5.4.1. Caveolin

Caveolin proteins 1–3 (Cav1-3) are located in PM invaginations, the so-called caveolae, which are enriched in cholesterol and sphingolipids (Figure 4 and Figure 5) [163,165,263,264]. They are essential in signal transduction and the coordination of signaling mechanisms within membranes. In particular, Cav1, which is incorporated into the membrane-like a hairpin, was reported to associate with Orai1, thereby regulating SOCE activation. It was suggested that Cav1 has a higher affinity for part of the TM4 domain (aa 250–258) as well as the N-terminus (aa 52–60) of Orai1. These interactions were found to be responsible for Cav-1 mediated internalization of Orai1 [161]. Based on this interaction, it is also likely that Cav-1 is vital to move Orai1 into cholesterol- and sphingolipid-rich regions [122,162]. In line with these findings, overexpression or knock-down of Cav-1 enhanced or decreased store-operated Ca^2+^ currents along with the number of caveolae [163,164]. Consistently, downregulation or overexpression of Cav-1 reduced or enhanced Orai1 expression in the airway smooth muscle cells and, consequently SOCE. This regulatory role of Cav-1 occurred independently of STIM1 [165]. Herewith, Cav-1 is involved in the formation of signaling hubs, that are essential for downstream signaling, such as the activation of transcription factors [122,162,164]. Notably, coupling of Cav-1 with the CRAC channel complex selectively controls store-operated Ca^2+^ influx mediated downstream NFAT- or c-fos-signaling. Indeed, phosphorylation of Cav-1 Y14 interferes with c-fos activation, while NFAT signaling or Orai1 activation remained unaffected. This suggests that Cav-1 potentially controls different PM microdomains in a target-specific manner to selectively activate different downstream signaling pathways [164]. In contrast to the results in the smooth muscle cells of the respiratory system, these results could not be reproduced in other cell types. Alternatively, it was proposed that Cav-1 stabilizes the STIM1/Orai1 complex, which in consequence promotes SOCE [164] (Figure 4 and Figure 5, Table 1).

#### 5.4.2. Junctate and Junctophilin-4

Junctate is a TM-protein located in the ER membrane that contains a long ER-luminal C-terminus with a Ca^2+^ sensitive portion and a short cytosolic N-terminus (Figure 4). It was shown to contribute to the development of ER-PM contact sites, as the number and size of the latter are enhanced or decreased upon overexpression or downregulation of junctate, respectively [265]. Among a variety of Ca^2+^ ion channels, it can also form a macromolecular complex with the STIM1/Orai1 assembly (Figure 5, Table 1). Removal of the Ca^2+^ binding site enhanced cytosolic Ca^2+^ levels and facilitated STIM1 cluster formation. Junctate-mediated STIM1 clustering occurs in a PIP_2_ independent manner via their direct binding. From these results, it was concluded that STIM1-Orai1 activation can be either triggered directly via PIP_2_ in the PM, by direct interaction with Orai1 or depending on junctate which interacts with STIM1. The proposed mechanism is that junctate is located with Ca^2+^ bound at the ER-PM contact sites under resting cell conditions, potentially in complex with other lipid-binding proteins. Upon store-depletion, STIM1 moves to the ER-PM junctions with the support of junctate to eventually couple to Orai1, which is accompanied by the anchoring of the STIM1 C-terminus to PIP_2_ in the PM [166] (Figure 5, Table 1).

An additional junctional protein, junctophilin 4, one of the four members of the junctophilin family, anchored in the ER membrane and located at ER-PM contact sites, was reported to support the interplay of STIM1 and Orai1 in a PIP_2_-independent manner (Figure 4 and Figure 5). Downregulation of junctophilin-4 reduced ER Ca^2+^ levels and store-operated Ca^2+^ entry. Furthermore, translocation of NFAT to the nucleus and extracellular signaling-related kinase (ERK) pathways as well as the expression of cytokines and activation markers were abolished. The cytosolic domain of junctophilin-4 was demonstrated to physically interact with STIM1 to support its recruitment to ER-PM junctions. Indeed, a cytosolic fragment of junctophilin-4 impaired SOCE. In addition, junctate formed a complex with junctophilin-4 that synergistically coupled to STIM1 to support its recruitment to ER-PM junctions to sustain ER-Ca^2+^ homeostasis and stimulate store-operated Ca^2+^ influx [167].

#### 5.4.3. SARAF

SARAF is a single TM protein located in the ER with the N-terminus facing the ER lumen and the C-terminus residing in the cytosol (Figure 4, Table 1). It associates with STIM1 already under resting conditions at ER-PM junctions to inhibit spontaneous STIM1 activation. While the N-terminus of SARAF determines its activity, its C-terminus is responsible for direct interaction with STIM1 (Figure 5). Furthermore, SARAF alleviates the restoration of the resting, deoligomerized state of STIM1 upon Ca^2+^ store-repletion which is accompanied by slow Ca^2+^ dependent inactivation (SCDI) [141,168,169]. The critical SARAF binding site in STIM1 is located downstream of the SOAR domain and is called COOH-terminal inhibitory domain (CTID). Their co-regulation is established by a complex interaction of SARAF with different segments of the CTID [168]. SARAF-mediated SCDI of STIM1-Orai1 currents is diminished when the lysine-rich region at the very end of STIM1 C-terminus is deleted, PIP_2_ is depleted or when STIM1 is translocated to PIP_2_-poor regions. The interplay between STIM1 and SARAF is further facilitated by E-Syt1 and septin4, which retain STIM1 and Orai1 in PIP_2_-rich regions [141]. It is assumed that transient shuttling between PIP_2_-poor and -rich areas is an essential regulator of the STIM1/Orai1 machinery. Ca^2+^ store-depletion triggers STIM1/Orai1 coupling in PIP_2_-poor regions. Because PIP_2_ levels are low, SARAF is unable to interact with STIM1, and Ca^2+^ entry can occur to maximal levels. Upon translocation to PIP_2_-rich regions, the STIM1-Orai1 complex comes in direct contact with SARAF, which triggers SCDI and reduction of Ca^2+^ currents [141]. Overall, these results support previous findings that the store-operated activation of STIM1 and Orai1 is positively modulated by PIP_2_ [129].

#### 5.4.4. Septin

Septins are GTP-binding proteins that assemble into highly ordered structures, forming filaments that function as barriers. However, depending on the type of cell and septin, their function can vary substantially. Based on their differences in sequence, they can be grouped into four subfamilies. Septins form linear filaments in which the different variants occupy distinct positions (Figure 4, Table 1).

They are located in both the ER and the PM. They contain a polybasic domain that enables them to interact with phosphoinositides such as PIP_2_. Subsequent to store-depletion, septins dynamically rearrange from a relatively homogenous distribution in the PM to aggregates. They accumulate at ER-PM junctions but are spatially separated from the STIM1/Orai1 complexes. Nevertheless, they indirectly impact CRAC channel function in various manners depending on the septin type [172] (Figure 5, Table 1). Knock-down studies revealed that different septin homologues are involved in the modulation of CRAC channel function, including the control of the degree of STIM1-Orai1 clustering therein.

Septins 2, 4, and 5 are positive modulators of CRAC channels. They facilitate the accumulation of STIM1 at the ER-PM junctions and STIM1-Orai1 complex formation after store-depletion. Silencing of the three septins altered the arrangement of Orai1 channels which was in accord with a changed distribution of PIP_2_ in the PM and in consequence Orai1 activation [170]. Knockdown of one or more of these septin types is thought to destabilize septin filaments and thus their ability to maintain PIP_2_ organization, which is essential for intact interplay between STIM1 and Orai1. The polybasic region at the very end of STIM1 C-terminus interacts with the charged phospholipid headgroups and stabilize STIM1 in the extended state when coupled to Orai1. Indeed, knock-down of Septin4/5 reduced the density of STIM1 puncta [172]. Moreover, septins can hinder the pre-aggregation of Orai1 in resting cells, potentially through their ability to alter the allocation of lipids in the PM [170]. However, higher resolution studies are still required for a more detailed understanding of the role of septins in Orai1 organization and activation [170,171].

In addition, Katz and co-workers [172] suggested that septin 4/5 control the number of ER-PM junctions to indirectly promote STIM1-Orai1 coupling within these areas. Septin 4 regions are clearly separated from ER-PM junctions. Moreover, altered expression of septin 4/5 did not alter the diffusion characteristics of STIM1 under resting and store-depleted cellular conditions. Septins 4/5 never co-localize with Orai1, whether it is in a resting state or activated. Rather, it is STIM1 that recruits Orai1 to the ER-PM junctions. Subsequent to store-depletion, delayed diffusion of Orai1 was observed when separated from STIM1, which in turn is advantageous for establishing the resting state and STIM1-Orai1 rebinding. In this way, septins interplay with STIM1 and Orai1 in an indirect manner via lipids and proteins in the surrounding and are located in favorable positions to orchestrate Ca^2+^ signaling with a restructuring of cellular architecture [172]. Recently, PIP_2_ and septin4 were reported to affect remodeling of the actin cytoskeleton which is required to promote the clustering of STIM1 and Orai1 in the ER-PM junction [266].

In contrast to the positive modulatory effect of the above mentioned septins, septin 7 was reported to suppress Orai-mediated currents in Drosophila neurons [173], human neurons, and human neural progenitor cells (hNPCs) [174]. While reduction of SEPT7 leads to enhanced Orai1-dependent Ca^2+^ entry, septin 7 overexpression inhibited Orai1 currents. It is currently unclear how septins of different subgroups, which are still able to interplay with each other and form complexes, can have antagonistic rather than synergistic effects [171,173]. Additionally, a recent publication revealed that in humans septin 7 is essential for Orai1-mediated Ca^2+^ influx in progenitor cells of the neuronal system and differentiated neurons. It is proposed that the underlying mechanism for the action of human septin 7 is distinct from that of the Drosophila variant. While in the latter the GTPase domain is essential for the action of septin 7 on Orai1 channels, in the human variant a polybasic segment in the N-terminus of septin 7 exerts the regulatory function on Orai1 [174].

#### 5.4.5. STIMATE/TMEM110

STIMATE (STIM-activating enhancer, encoded by TMEM110), an ER-resident multi-transmembrane protein with a polybasic tail at the end of the C-terminus, is a positive regulator of the Ca^2+^ entry. It stabilizes ER–PM junctions required for the STIM–ORAI interplay (Figure 4 and Figure 5, Table 1). Down-regulation of STIMATE resulted in a significant reduction in STIM1 clusters at the ER-PM contact sites, greatly reduced store-operated Ca^2+^ entry, and reduced Ca^2+^-dependent gene transcription (NFAT). A 10% reduction in ER-PM junction formation after STIMATE elimination can only partially explain this effect. More importantly, STIMATE physically interacts with STIM1-CC1 to facilitate the conformational switch of STIM1 to the active state [175].

#### 5.4.6. RASSF4

RASSF4 belongs to the family of RAS association domains, which contain a RAS (RA) association region and a Salvador–RASSF–Hippo (SARAH) domain in their C-termini [267]. Predictions suggest that the RA domain of RASSF4 interacts with small G proteins [268]. The SARAH domain can trigger protein interaction and the formation of dimers [269,270,271]. Altered expression levels of RASSF4 is linked to cancer development and inhibition of the Hippo pathway [176], which is known to regulate organ size by controlling cell proliferation and apoptosis [272,273]. In recent years, RASSF4 was shown to play a fundamental role in Ca^2+^ signaling and the formation of functional CRAC channels. It controls ER-PM contact site formation, E-Syt2- and E-Syt3 mediated tethering of ER-PM regions, and PIP_2_ status in the PM (Figure 4, Table 1). Suppression of RASSF4 impaired all these events and additionally reduced translocation of STIM1 to ER-PM junctions and consequently, store-operated Ca^2+^ entry. PIP_2_ is generated from PI4P by PIP5K. An upstream regulator of PIP5Ks and PIP_2_ is adenosine diphosphate ribosylation factor 6 (ARF6), a small G protein. RASSF4 regulates the activity of ARF6 through interaction. Downregulation of RASSF4 reduced the activity of ARF6 and in consequence the PIP5K-mediated production of PIP_2_ from PI4P [176].

#### 5.4.7. Adenylyl Cyclase 8 (AC8)

The function of the STIM1/Orai1 complex is further regulated by the adenylyl cyclase (AC) 8. In the family of adenylyl cyclases, it is one of those regulated by Ca^2+^ and binds to cholesterol-rich regions in the PM [274]. This membrane association can be disrupted by cyclodextrin-induced cholesterol degradation, which in turn negatively affects SOCE [275]. Mechanistically, AC8 processing, targeting, and responsiveness are influenced by a dynamic interaction with Cav-1 [276,277] (Figure 4, Table 1). The interaction of AC8 at the PM is enhanced by the cytoskeleton of the cell as well as an interaction with cholesterol. This suggests that AC8 migrates along the cortical actin to cholesterol-rich areas, where the produced cAMP regulates its interaction with the actin cytoskeleton [278]. The response of AC8 to SOCE is regulated by its direct association with AKAP79 (A-kinase anchoring protein). AKAP79 is specifically bound to phospholipids in cholesterol-rich regions of the PM by three N-terminal polybasic regions and by specific palmitoylation/S-acetylation of certain cysteine residues. This enables the modulation of the AC8 activity in a SOCE dependent manner and consequently PKA-mediated phosphorylation of proteins residing in cholesterol-rich regions, including AC8 [177,178]. In addition, the targeting of AC8 by AKAP79 to the PM was shown in HEK 293 cells to allow direct coupling to Orai1, where they modulate the activity of each other [279,280]. Recently, however AC8 was reported to be expressed in neuronal cells, but not in HEK 293 and various immune cells. Alternatively, AKAP79 alone, bound directly to Orai1, was demonstrated to control Ca^2^^+^-calcineurin and cAMP-protein kinase signaling pathways independently of AC8 in separate nanodomains [281]. AKAP79 was identified to associate with the N-terminal segment aa 39–59 of Orai1 which allows upon local Ca^2+^ entry through Orai1 the activation of associated calcineurin and rapid translocation of NFAT into the nucleus [282]. 

## 6. Lipid-Mediated Modulation of the Co-Regulation of CRAC Channel Components with Other Ion Channels

Dynamic modulation of the lipid environment at ER-PM contact sites controls not only the sole interplay of STIM1 and Orai1 but also their co-regulation with other ion channels.

A prominent example is that cholesterol-rich regions determine the coregulation of TRPC1, STIM1, and Orai1. Indeed, chemical depletion of cholesterol interfered with the store-depletion induced interaction of STIM1 with Orai1 as well as TRPC1 [283]. Cholesterol-rich regions were demonstrated to be essential for the initial phase of SOCE activation but not the maintenance of SOCE involving these three transmembrane proteins. After full recruitment of the three TM proteins, both the channel complex as well as SOCE remained unaffected by cholesterol depletion [134].

Both Orai1 and TRPC1 channels contain a Cav-1 binding site, which may be responsible for their targeting to caveolae- and cholesterol-rich regions [161,165,284]. Indeed, down-regulation of Cav-1 weakened SOCE in cells containing STIM1, Orai1 and TRPC1 [263,285]. Meanwhile, the co-regulation of Cav-1, TRPC1, and STIM1 was explored in detail. Notably, though TRPC1 is able to interact with Cav-1 [263,284,285,286,287], TRPC1 is activated by STIM1 once it has translocated to cholesterol-rich regions [135,288]. Indeed, the interaction between STIM1 and TRPC1 in cholesterol-rich regions enables store-operated TRPC1 activation, whereas outside these areas and without binding to STIM1, TRPC1 acts as an agonist-activated channel [288]. In line with these findings, TRPC1-SOCE was shown to trigger the disassembly of the TRPC1-Cav1 complex in the presence of STIM1 [289], thus explaining the previously observed scenario of STIM1-TRPC1 interaction in cholesterol-rich regions [135]. Nevertheless, the extent of STIM1-TRPC1 interaction is defined by Cav-1, as its knock-down impaired store-operated STIM1-TRPC1 interaction and localization in Cav-1-rich microdomains, although the STIM1/Orai1 interaction remained unaffected [286]. Overall, Cav-1 is required to trigger translocation of TRPC1 to cholesterol-rich regions. This is an indispensable prerequisite for STIM1-TRPC1 coupling event in these membrane regions, where Cav-1 is assumed to finally dissociate.

Another example for ion channel co-regulation in cholesterol-rich regions represents the interplay of Orai1 with the Ca^2+^ activated K^+^ ion channel, SK3, primarily in cancer cells. In contrast to healthy human breast, colon and prostate tissue cells [290,291,292,293,294,295], which only express Orai1, analogous cancer cells, co-express SK3 and Orai1 [292,293,294,296,297,298].

In breast cancer cells, the SK3-Orai1 co-regulation was reported to result in constitutive Orai1 dependent but STIM1 independent Ca^2+^ influx. The reasons for these effects are likely due to their close co-localization in cholesterol-rich regions as well as hyperpolarization due to Ca^2+^-mediated SK3 channel activation. In fact, only silencing of Orai1, but not STIM1, abolished Ca^2+^ entry [290,291,292,293,294,295,299]. SK3/Orai1 co-regulation is responsible for breast cancer cell proliferation and migration or can even trigger bone metastasis [292,293,294,296,297,298]. Similarly, colon cancer cell growth is driven by an interplay of SK3 and Orai1 which is further supported by STIM1 and TRPC1 [292]. We provided recently further indications that the SK3/Orai1 interplay also occurs in LNCaP cells and drives their proliferation [300]. Moreover, we discovered that SK3/Orai1 co-regulation can also occur in the standard overexpression cell line, HEK 293. In line with the findings in cancer cells, we discovered a close co-localization of SK3 and Orai1, which is likely responsible for local enhancements in Ca^2+^ levels. Local Ca^2+^ elevations are in turn able to strongly boost SK3 K^+^ currents compared to cells only containing SK3. In addition, the Orai1 E106Q pore mutant, removal of extracellular Ca^2+^, using an extracellular Na^+^ divalent free solution, enhancing intracellular Ca^2+^ buffering, or the application of Orai1 channel blockers reduced the extent of Orai1-mediated boost in SK3 K^+^ currents, highlighting that the observed effects are specific for Orai1. We further discovered that one critical determinant for this co-regulation is the CaM binding site in SK3, because Orai1 overexpression decreased the interaction of SK3 with CaM as well as the CaM_MUT_s. Moreover, SK3 K^+^ currents abolished by the CaM_MUT_ were partially restored by co-expression of Orai1. At the level of Orai1, an intact pore and both cytosolic strands, except the first 26 aa, are indispensable for the co-regulation of SK3 and Orai1 [112,301].

In breast and colon cancer cells, this SK3/Orai1 interplay was shown to occur in cholesterol-rich regions. Their co-regulation was disrupted by the application of the alkyl ether lipids edelfosine and ohmline, which are selective SK3 inhibitors. Specifically, these lipids hampered the co-localization of SK3 and Orai1, displaced Orai1 out of the cholesterol-rich regions and abolished constitutive Ca^2+^ entry [293,302,303,304]. Furthermore, Cav-1 was reported to further contribute to STIM1/TRPC1/Orai1/SK3 complex formation in colon cancer [292]. Overall, there is clear evidence that SK3 and Orai1 interplay with each other. However, since the two ion channels do not interact directly, the direct connection partners for this co-regulation remain to be identified.

One such direct interaction partner could be the stress-activated chaperone, SigmaR1. It was reported to interfere with the CRAC channel machinery [305] or to boost the SK3/Orai1 interplay [294]. In HEK293 cells, SigmaR1 interacted with STIM1 and moved into ER-PM junctions, slowing the interaction of STIM1 with Orai1, and leading to reduced STIM1/Orai1 currents [305]. In contrast, SigmaR1 was directly bound to SK3 in breast and colon cancer cells and supported the interplay of SK3 and Orai1 [294]. In fact, knockdown of SigmaR1 or use of the ligand igmesine, which inhibits SigmaR1, abolished the SK3/Orai1 interplay of channels in cholesterol-rich nanodomains in breast and colon cancer cells [294]. However, it remained unclear whether the direct interaction of SigmaR1 and SK3 enhances the colocalization with Orai1. A possible approach would be to test whether SigmaR1 also interacts with Orai1. The different ways of SigmaR1-mediated regulation of Ca^2+^ sensitive proteins STIM1 and SK3 were explained by the fact that only the respective cancer cells, but not the HEK293 cells, express SK3. To prove this hypothesis, it may be worthwhile to investigate how SigmaR1 affects the coregulation of STIM1, Orai1, and SK3 co-expressed in HEK 293 cells, compared to only co-expression of either STIM1 and Orai1 or Orai1 and SK3.

## 7. Conclusions and Perspectives

It is now very clear that the CRAC channel machinery is not only composed of STIM and Orai proteins, but in particular the function and co-regulation of STIM1 and Orai1 are fine-tuned by a variety of proteins as well as lipids either through direct interaction or indirectly by modulating the properties of ER-PM contact sites. It is not yet explored whether the function of other STIM and Orai isoforms can be also modulated by lipids.

Phospholipids, especially PIP_2_, are involved in the formation of a stable STIM1/Orai1 complex. In this context, the STIM1-PIP_2_ binding site in particular is well characterized, whereas for Orai1 the exact mechanism of PIP_2_ regulation is not yet known. Furthermore, cholesterol, sphingomyelin as well as other phospholipids play important roles in the STIM1/Orai1 activation process. However, current studies on the modulatory role of cholesterol are controversial, for sphingomyelin the mechanism of CRAC channel modulation is unknown, and it is still unclear whether and which phospholipids in the ER membrane maintain STIM1 in the currently postulated closed conformation.

Since lipid–protein interactions are complex, it is a non-trivial task to elucidate them using standard structural biology and fluorescence methods. Although lipid–protein interactions are of immense importance, functionally relevant interactions with lipids have only been structurally proven for 10% of membrane proteins. This is partly because lipid interactions are highly dynamic. In addition, weakly bound lipids are often lost during purification steps for structural analysis. In conventional microscopy, the diffraction barrier of 200 nm is limiting in the resolution of lipid-protein interactions. Novel methods, such as purification of proteins in a membrane-like environment, as well as high-resolution microscopy techniques, such as the PALM and stimulated emission depletion microscopy (STED) [306] could be groundbreaking for a better understanding of the direct interaction between CRAC channels and lipids.

Currently, a variety of accessory proteins are known to modulate the ER-PM junctions and, consequently, the co-regulation of STIM1 and Orai1. On the one hand, some of them interact directly with STIM1 or Orai1 to control their interplay, and on the other hand others are involved in the non-vesicular lipid transport at ER-PM contact sites to modulate CRAC channel activation. For some of these proteins localized at the ER-PM junctions it is not yet clear whether they form a direct interaction with either CRAC channel component. Although fragments of the STIM1/Orai1 interplay with accessory proteins are often well-known, the complete molecular picture of all interactions is still lacking. Furthermore, it remains unclear whether all these proteins interplay with each other in the same cell at the ER-PM junction or whether this diversity of available ER-PM proteins act in a cell type-specific manner, as it was identified for E-Syts [136,143]. Some proteins may also form ER-PM junctions independently of STIM proteins, opening up the intriguing possibility that different types of junctions form in a cell in response to stimulation by varying amounts of agonist.

Overall, an improved and detailed understanding of the modulatory roles of lipids and lipid-dependent factors on the CRAC channel machinery would enlarge our knowledge of physiological and pathophysiological downstream signaling processes. The complexity with which lipids influence the CRAC channel function may allow an enhanced fine-tuning of target-specific drugs. Currently, several CRAC channel drugs are available, however, only a few of those have reached clinical trials due to low selectivity or unwanted side effects [307,308]. Thus, a better understanding of the formation and composition of the ER-PM junctions would enable the development of drugs that target new sites or distinct steps in the process with greater precision.

## Figures and Tables

**Figure 1 biomolecules-12-00352-f001:**
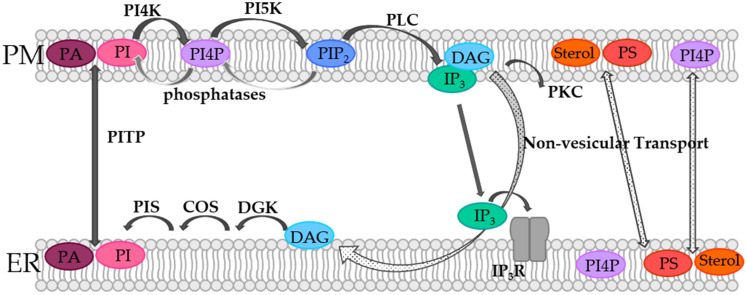
Lipid transport and phosphoinositide (PI) cycle at the ER-PM contact sites. PI is consecutively transformed at the PM into PI4P by PIP4K and PIP_2_ by PIP5K (PIP_2_ can be converted to PI4P or PI by phosphatases). PM-receptor stimulation can trigger PLC mediated hydrolysis of PIP_2_ to form DAG and IP_3_. While DAG can activate PKC or some ion channels, IP_3_ activates the ER located IP_3_R. Non-vesicular transport mechanisms (see Section 5) can transport DAG to the ER membrane, where it is converted to PI. PITPs transport PI back to the plasma membrane. Furthermore, sterols, PS, and PI4P are transported between the two membranes via non-vesicular transport (see Section 5).

**Figure 2 biomolecules-12-00352-f002:**
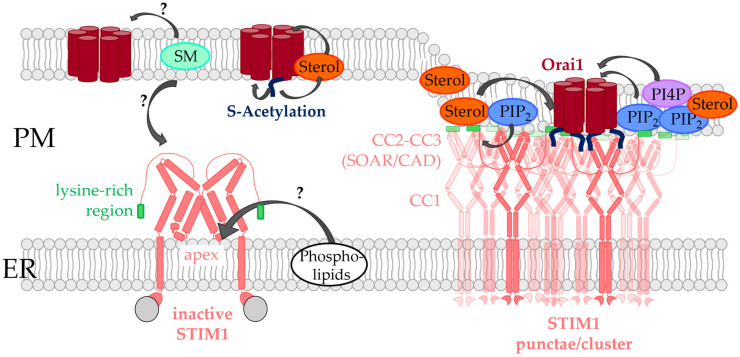
STIM1, Orai1, and their interplay with lipids. In the resting state, STIM1, located in the ER membrane, captures a quiescent and folded state, while Orai1, located in the PM, possesses a closed pore. After store-depletion STIM1 undergoes a conformational change, oligomerizes, and couples to Orai1. Although the interaction and function of the two proteins is sufficient for CRAC channel activation, their machinery is modulated by a variety of lipids including PIP_2_, PI4P, cholesterol, sphingomyelin and possibly also ER-phospholipids as outlined in detail in Section 4.

**Figure 3 biomolecules-12-00352-f003:**
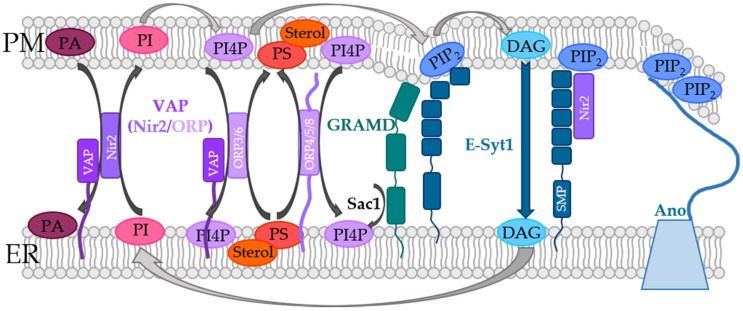
ER-PM spanning proteins involved in the modulation of the CRAC channel complex. In addition to the PI cycle, phospholipids can also be transported by ER-PM spanning protein, allowing non-vesicular transport. The localization of ER-PM spanning proteins depends mainly on the lipid composition in the PM, in particular the PIP_2_ levels. E-Syt proteins are anchored in the ER membrane and are either Ca^2+^-dependently (E-Syt1) or -independently (E-Syt2, E-Syt3) bound to PIP_2_ in the PM. E-Syt proteins can transport DAG from the PM to the ER membrane. Furthermore, E-Syt1 co-localizes with Nir2 to modulate PIP_2_ levels in the PM. VAP proteins exchange PI and PA between the ER and PM. Moreover, VAP proteins associate with Nir2 or ORP3/6 to span from the ER to the PM. Distinct ORP variants, ORP5 and ORP8, reside in the ER and bridge the distance to the PM without additional proteins. VAP/ORP proteins and complexes exchange phosphatidylserine (PS) and sterols in the ER for PI4P in the PM. ORP proteins supply PI4P to Sac1 phosphatase in the ER membrane. GRAMD proteins contain a single TM domain located in the ER, bind to PIP_2_ in the PM, and are involved in the transport of sterols. ANO8 is multi-transmembrane domain protein in the ER that binds PIP_2_ in the PM.

**Figure 4 biomolecules-12-00352-f004:**
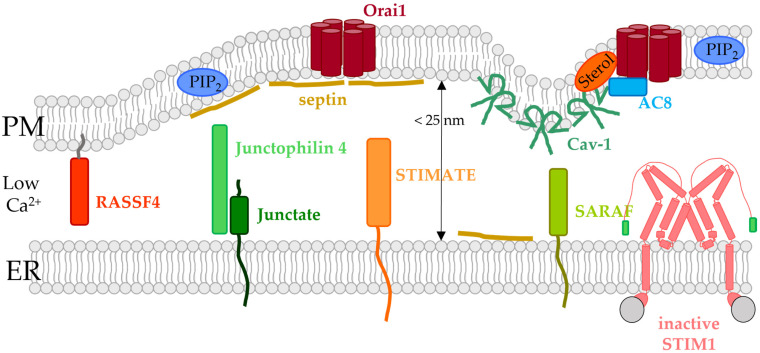
Modulatory proteins at the ER-PM contact sites involved in modulating the interplay of STIM1/Orai1. Schematic of all modulatory proteins located at the ER-PM contact sites reported to be critical for modulation of the STIM1/Orai1 complex. Junctate, junctophilin, STIMATE, and SARAF are located in the ER. RASSF4 and septins are located close to and Cav1 is anchored within the PM.

**Figure 5 biomolecules-12-00352-f005:**
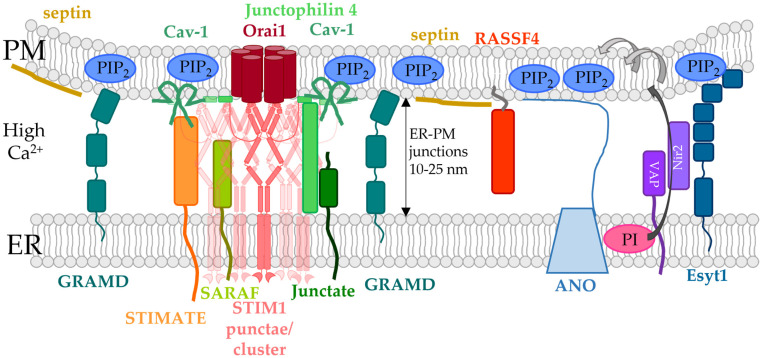
Complex interplay of ER-PM proteins and lipids with the STIM1/Orai1 channel. GRAMD, Ano8 and E-Syt proteins modulate the ER-PM contact site to facilitate the interplay of STIM1 and Orai1 in an indirect manner. STIMATE, junctate, and junctophilin 4, together with PIP_2_, directly interact with STIM1 to support its oligomerization, association near the PM, and coupling with Orai1. SARAF contributes to efficient restorage of the closed state of STIM1 through direct interaction. RASSF4 and septins modulate PIP_2_ levels in the PM and are thus indirectly involved in STIM1 activation. Cav-1 is thought to move Orai1 into cholesterol-rich regions.

**Table 1 biomolecules-12-00352-t001:** List of lipids and ER-PM junctional proteins that modulate the function of STIM1 and/or Orai1. The lipids and ER-PM junctional proteins are listed together with their location in the cell critical for the modulatory effect on STIM1 and/or Orai1, their effects on the functions of the CRAC channel, STIM1, and Orai1, whether they interact directly with STIM1 or Orai1 and the corresponding references.

**Lipid Type**	**Location**	**Effect on Endogenous CRAC**	**Effect on STIM1**	**Effect on Orai1**	**Direct Interaction with STIM1 or Orai1**	**Ref.**
PIP_2_	PM	↑	↑ STIM1 assembly at PIP_2_ rich region	↑	STIM1 K-rich region (aa 672–685) Orai1 N-terminus (aa 28–33)	[129,130]
PI4P	PM	↑	↑	n.d.	n.d.	[130,131,132,133]
cholesterol	PM	↑	↓	↓	STIM1 OASF (I364) Orai1 N-terminus (L74, Y80)	[119,120,121,122,129,134,135]
sphingo-myelin	PM	↑	n.d.	n.d.	n.d.	[128]
ER-phospho-lipids	ER	n.d.	↓	not applicable	STIM1 apex (F394K/D)	[83]
S-acylation (post-transla-tional tethering of fatty acids)	PM	↑	not applicable	↑	Orai1 C143	[123]
**Protein within ER-PM junction**	**Location**	**Effect on endogenous CRAC**	**Effect on STIM1**	**Effect on Orai1**	**Direct interaction with STIM1 or Orai1**	
E-Syt	ER-PM junction (E-Syt1 located there in a Ca^2+^ dependent manner) embedded in ER	no/minor effect (HeLa cells)	n.d.	n. d.	indirect	[136,137,138,139,140,141,142]
E-Syt1	$ER\overset{{\left[Ca^{2+}\right]}_{cyt}↑}{\to }$ ER−PM Rearrangement of ER-PM contact sites	no/minor effect	supports repletion of ER stores with Ca^2+^ (HEK293 cells)	n.d.	indirect	[142]
E-Syt2	ER-PM junction	↑ (E-Syt2S)	↑ (E-Syt2S) ↓ (E-Syt2L)	n.d.	indirect	[143]
E-Syt3	ER-PM junction	No effect/ minor (when knocked-out together with E-Syt1/2)	n.d.	n.d.	indirect	[136]
E-Syt1 + Nir2	ER-PM junction	↑	↑ Co-localization with STIM1	n.d.	Indirect dependent on DAG levels in the membrane	[140,144,145,146]
VAP	$ER\overset{FFAT}{\to }$ ER−PM	n.d.	n.d.	n.d.	indirect	[147,148,149]
VAP + ORP3/6	ER-PM junction	↓	↓	n.d.	n.d.	[150,151,152]
ORP5/8	ER-PM junction, ER-Mito	n.d.	n.d.	n.d.	n.d.	[148,150,152,153,154]
GRAMD2A	ER-PM junction	not required (more studies required)	↑ Supports STIM1 translocation STIM1 puncta formation	n.d.	Indirect Co-localizes with E-Syt and STIM1 (PIP_2_ required)	[155,156,157,158,159]
ANO8	ER-PM junction	↑	↑ STIM1 oligomeri-zation/clustering	↑ STIM1/Orai1 interaction; inactivation	Indirect (further studies necessary)	[160]
Caveolin-1	PM	↑	independent of STIM1	↑ STIM1/Orai1 coupling	Direct with Orai1 (aa 52–60; aa 250–258)	[122,161,162,163,164,165]
Junctate	ER	↑	↑	↑	Direct with STIM1 (STIM1 N-terminus)	[166]
Juncto-philin 4	ER	↑	↑ supports recruitment of STIM1 to ER-PM	n.d.	Direct interaction with STIM1	[167]
Junctate + juncto-philin 4	ER-PM	↑	supports recruitment of STIM1 to ER-PM	n.d.	Direct interaction with STIM1	[166,167]
SARAF	ER	↓	↓	↓	STIM1 CTID	[141,168,169]
Septin	PM	depending on the septin type			indirect	[170,171,172,173,174]
STIMATE	ER	↑	↑	↑	STIM1-CC1	[175]
RASSF	PM	↑	↑	↑	indirect (modulates PIP_2_ levels)	[176]
AC8	PM	↑ (Regulated by its direct association with AKAP79)	↑	↑	Orai1	[177,178]

## Data Availability

No new data were created or analyzed in this study. Data sharing is not applicable to this article.

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
