# Peer review of "The Role of Lipids in CRAC Channel Function"

_biomolecules, 2022, doi:10.3390/biom12030352_

Round 1

Reviewer 1 Report

Comments and edits have been made directly on the PDF. 

Author Response

We thank the reviewer for constructive comments.

As suggested, we adapted the manuscript accordingly.

We included that lipids can also affect trafficking of ion channels (line 76).

We apologize for the misunderstanding in Figure 2: The potential phospholipid mediated regulation of STIM1 refers to the apex, as described in section 3.5. Figure 2 is now accordingly adapted. There is no evidence for the maintenance of the closed state of STIM1 by co-regulation of the polybasic region and ER phospholipids.

As suggested, we moved the sentence at the end of the first sentence of this paragraph: “These exhibit a folded configuration in which CC2 and CC3 resemble the capital letter “R” and form an antiparallel V-shape in the dimeric state [72]. In the quiescent state, this V-shape is upside down and faces the ER membrane.” Moreover, we clearly stated that the folded state corresponds to the inactive/quiescent state (line 181-188).

As suggested, we refer to the Gill lab manuscript (Nat Comm 2014; 5:3183). However, as our review focuses on STIM1 and Orai1, and other isoforms have not been explored in terms of lipid-mediated regulation, we have decided not to elaborate on F394 in STIM1 and the analogous position in STIM2.

As suggested, we adapted: “After deletion of this polybasic motif, ER depletion-induced translocation of STIM1 to the PM is impaired, even though STIM1 retains its ability to form oligomers [129].” (line 277)

We described briefly the finding in the two references 108 and 104 (originally 93 and 97) (line 290).

We further mentioned that MßCD has been shown to decrease membrane localization of STIM1 and disrupt its interaction with Orai1 (line 328).

We adapted the sentences at the beginning of section 4.4.

We referenced the recent paper from the Ambudkar lab showing that both PIP2 and septin4 affect remodelling of the actin cytoskeleton (line 885-887).

We also included the recent paper of Parekh lab (line 946-949).

As suggested, we mentioned that also other STIM and Orai isoforms contribute to CRAC channels. However, in this review we focused on STIM1 and Orai1, since the lipid mediated regulation of other isoforms is still unexplored (see section 2., 1st paragraph and 6., 1st paragraph).

Finally, we also adapted the section 6. (discussion and perspectives) according to the reviewers suggestions.

Reviewer 2 Report

In the current review Maltan and colleagues present a comprehensive summary of how store operated Ca2+ entry is regulated by lipids of the cell membranes. The introduction provides a very satisfactory overview of main lipid components of membranes involved in SOCE namely PM and ER. The authors then spared no effort in detailed description of lipids that directly or indirectly regulate SOCE where they introduced an increasingly interesting subject of membrane contact sites with a very comprehensive overview of SOCE-regulatory proteins. Within the different sections, the authors proposed hypotheses based on the presented literature and suggest further studies in a logic and inspiring manner.

One concern of mine is that although the lipid regulation was extensively studied for Orai1-STIM1, the major component of ICRAC channels, it is worth mentioning at least in the introduction that there are other isoforms (Orai2, 3 and STIM2) that contribute, in a cell type-extent, to store operated Ca entry, in addition to their proposed store independent activity (Orai3 as part of ARC channel). If and how these channels are regulated by lipids is, to my knowledge, still unexplored.

Minor comments:

Line 38 please replace: presence of “double bonds” with saturation state

In section line 58 to 76: would be interesting to add how lipid (dys)regulation of ion channels is involved in development and/or progression of diseases, e.g cardiovascular diseases and K+ channels

Line 80: I suggest to modify the sentence (this work…) to: next section discusses lipid composition and dynamics of the ER and the PM

Line 108: please delete “some”

Figure 1: while introducing the PI cycle, maybe it is worth indicating that PIP4K and PIP5K have counter partner phosphatases the activity of which affects the dynamics and composition of lipids in the PM. The authors mentioned phosphatases in the text (line 128)

Line 149: please replace contact with proximity (since contact per se can’t be close or far)

Line 157: please insert “in the form of dimer at rest”

In section 4.4.7 the authors refer to regulation of SOCE by AKAP79. Might also be useful to refer to this regulation in the earlier section 3.1 since higher levels of PIP2 have been shown to support recruitment of STIM and docking of AKAP79 to PM (Dell’Acqua et al., 1998).

Section 4.4 please check paragraph formatting.

Author Response

We thank the review for the positive evaluation of and the constructive comments to our review.

In the current review Maltan and colleagues present a comprehensive summary of how store operated Ca2+ entry is regulated by lipids of the cell membranes. The introduction provides a very satisfactory overview of main lipid components of membranes involved in SOCE namely PM and ER. The authors then spared no effort in detailed description of lipids that directly or indirectly regulate SOCE where they introduced an increasingly interesting subject of membrane contact sites with a very comprehensive overview of SOCE-regulatory proteins. Within the different sections, the authors proposed hypotheses based on the presented literature and suggest further studies in a logic and inspiring manner.

One concern of mine is that although the lipid regulation was extensively studied for Orai1-STIM1, the major component of ICRAC channels, it is worth mentioning at least in the introduction that there are other isoforms (Orai2, 3 and STIM2) that contribute, in a cell type-extent, to store operated Ca entry, in addition to their proposed store independent activity (Orai3 as part of ARC channel). If and how these channels are regulated by lipids is, to my knowledge, still unexplored.

As suggested, we mentioned that also other STIM and Orai isoforms contribute to CRAC channels. However, in this review we focused on STIM1 and Orai1, since the lipid mediated regulation of other isoforms is still unexplored (see section 2., 1st paragraph and 6., 1st paragraph).

Minor comments:

Line 38 please replace: presence of “double bonds” with saturation state

We replaced by “saturated state”

In section line 58 to 76: would be interesting to add how lipid (dys)regulation of ion channels is involved in development and/or progression of diseases, e.g cardiovascular diseases and K+ channels

We added a paragraph on lipid-(dys)regulation of ion channels in disease (line 79-88).

Line 80: I suggest to modify the sentence (this work…) to: next section discusses lipid composition and dynamics of the ER and the PM

We modified as suggested.

Line 108: please delete “some”

We deleted as suggested.

Figure 1: while introducing the PI cycle, maybe it is worth indicating that PIP4K and PIP5K have counter partner phosphatases the activity of which affects the dynamics and composition of lipids in the PM. The authors mentioned phosphatases in the text (line 128)

We further show counter-phosphatases in Figure 1 and mention examples in the text (line 141)

Line 149: please replace contact with proximity (since contact per se can’t be close or far)

We replaced it.

Line 157: please insert “in the form of dimer at rest”

It is exchanged.

In section 4.4.7 the authors refer to regulation of SOCE by AKAP79. Might also be useful to refer to this regulation in the earlier section 3.1 since higher levels of PIP2 have been shown to support recruitment of STIM and docking of AKAP79 to PM (Dell’Acqua et al., 1998).

Dell’Acqua et al., 1998 reports on AKAP79 binding to PIP2 in the plasma membrane. Since section 3.1 is focused on STIM1 and its regulation by PIP2 we decided not to mention AKAP79 in this section. But we referenced Dell’Acqua et al. in 4.4.7.

Section 4.4 please check paragraph formatting.

Checked.

Reviewer 3 Report

In the manuscript entitled “The role of lipids in CRAC channel function”, with a focus of lipids/proteins at ER-PM junctions, Maltan et al provided an up-to-date review on the regulation of CRAC channels. This review goes beyond lipids, provided a quite comprehensive review of CRAC channel regulation by proteins as well. It is already in a fine shape, and I only have one minor suggestion. If possible, it would be nice to add more details in the figures. Just name a few, show the polybasic region of STIM1, add symbols to show enhancement/inhibition on CRAC channels by lipids or regulatory proteins.

Author Response

We thank the review for the positive evaluation of and the constructive comments to our review.

As suggested, we provided now in Figure 2 more details on some of the STIM1 domains.

Instead of showing symbols in the figures for enhancement/inhibition of CRAC channel activity, we decided to make two tables listing lipids and ER-PM junction proteins and their effects on the CRAC channel, STIM1, Orai1 and whether there is a direct binding or not.